How much biomass do plant communities pack per unit volume?

Proulx Raphaël raphael.proulx@uqtr.ca
Rheault Guillaume
Bonin Laurianne
Roca Irene Torrecilla
Martin Charles A.
Desrochers Louis
Seiferling Ian
Canada Research Chair in Ecological Integrity, Département des Sciences de l’Environnement, Université du Québec à Trois-Rivières , Trois-Rivières, Québec , Canada
Lortie Christopher
Electronic publication date: 2015 Mar 19
Publication date: 2015
Volume: 3
Electronic Location ID: e849
Received 2014 Dec 8; Accepted 2015 Mar 3
Copyright: © 2015 Proulx et al.
Copyright year: 2015
Copyright holder: Proulx et al.
License: This is an open access article distributed under the terms of the Creative Commons Attribution License, which permits unrestricted use, distribution, reproduction and adaptation in any medium and for any purpose provided that it is properly attributed. For attribution, the original author(s), title, publication source (PeerJ) and either DOI or URL of the article must be cited.
License URL: https://creativecommons.org/licenses/by/4.0/

Keywords: Packing density, Biodiversity, Plant geometry, Ecosystem, Self-thinning, Species coexistence

Funding: Natural Sciences and Engineering Research Council of Canada This research was supported by a Natural Sciences and Engineering Research Council of Canada (NSERC) research grant to R Proulx. The funders had no role in study design, data collection and analysis, decision to publish, or preparation of the manuscript.

==============================
Aboveground production in terrestrial plant communities is commonly expressed in amount of carbon, or biomass, per unit surface. Alternatively, expressing production per unit volume allows the comparison of communities by their fundamental capacities in packing carbon. In this work we reanalyzed published data from more than 900 plant communities across nine ecosystems to show that standing dry biomass per unit volume (biomass packing) consistently averages around 1 kg/m3 and rarely exceeds 5 kg/m3 across ecosystem types. Furthermore, we examined how empirical relationships between aboveground production and plant species richness are modified when standing biomass is expressed per unit volume rather than surface. We propose that biomass packing emphasizes species coexistence mechanisms and may be an indicator of resource use efficiency in plant communities.

Introduction

Uncovering general principles that govern the functioning of ecosystems is a long standing objective of community ecology. One such principle, the self-thinning rule, predicts that the standing biomass of plant communities increases with decreasing stem density in crowded stands (Hutchings, 1979; Westoby, 1984). Weller (1989) proposed a simple geometric model of the self-thinning rule in which the standing biomass (g/m2) of different communities is negatively correlated to stem density. When expressed per unit volume (g/m3), however, his model also predicted that standing biomass should be independent of stem density across the plant kingdom (Weller, 1989).

The absence of a general relationship between standing biomass per unit volume and stem density across a range of ecosystems would have two important implications. First, it would evince that plants cannot allocate more carbon to aboveground compartments than imposed by competition for volumetric resources, such as light. Secondly, plant communities would differ from one another in their capacity to grow tall, yet compare in their capacity to pack carbon. Weller (1989) reported a positive relationship between dry biomass per unit volume and stem density, suggesting more efficient carbon packing in communities growing at a higher density. However, Weller’s relationship relied on two datasets, and thus remains inadequately tested.

Aboveground production, or yield, in terrestrial plant communities is commonly measured in amount of dry biomass per unit surface. Alternatively, expressing production per unit volume could better reflect how plants utilize all dimensions of the space in which they grow, hence emphasizing coexistence mechanisms. For instance, plants adapted to grow in the shade may contribute to biomass packing by filling the understorey space more efficiently (Claveau, Messier & Comeau, 2005; Valladares & Niinemets, 2008). Moreover, plants characterized by different growth rates or lifespans may increase biomass packing by acquiring resources on different spatial or time scales (Augspurger, 2008).

Biomass packing can be conceptualized as the amount of standing vegetation living within a sampling box. This box delineates a three-dimensional space, either physical or virtual, within which the vegetation is confined. Thus, biomass packing represents the amount of standing vegetation (expressed as dry biomass or occupied plant volume) scaled per unit of sampled volume.

The present study addresses two questions: Are there fundamental limits to the amount of standing dry biomass that is packed per unit volume in terrestrial plant communities? Is biomass packing affected by community composition and species richness? To investigate these questions, we reviewed the scientific literature and compiled data for more than 900 plant communities. This synthesis highlights broad patterns in biomass packing across contrasted ecosystem types. It is not a meta-analysis of the reviewed literature, which would require the inclusion of climatic or edaphic moderator variables as well as accounting for spatial auto-correlation and random (data source) effects.

Materials & Methods

We searched the scientific literature for studies reporting both dry standing biomass per unit surface and mean height of canopy plants in vegetation stands. The search targeted studies that monitored vegetation stands close to their peak of biomass production (Table 1). We converted all biomass values to g/m2 and height values to m (Table S1). In the literature, we found only Weller’s (1989) study reported direct measures of biomass per unit volume in terrestrial ecosystems. As such, Weller’s dataset was not used in our analysis, but as a benchmark to compare those results against. From all other aggregated data, we calculated the packing density of each plant community by dividing standing dry biomass per unit surface by community height.

Table 1 Ecosystems and plant communities (No. stands) from various sources reporting both total dry standing biomass per unit surface and mean height of canopy plants.

Ecosystem	No. stands	Location	Sources	
Cropland—Biofuel	19	USA	(Propheter et al., 2010)	
Cropland—Corn	13	USA	(Wilhelm et al., 2011)	
Forest—Tropical	19	Columbia, Cambodia, Thailand,
Venezuela, Malaysia, Borneo, Puerto Rico	(Yamakura et al., 1986; Weaver & Murphy, 1990)	
Forest—Pacific (west-coast)	45	Canada	(Gillis, Omule & Brierley, 2005)	
Forest—Temperate	174	USA, Canada	(Whittaker et al., 1974; Gillis, Omule & Brierley, 2005)	
Forest—Boreal	126	Canada	(Gillis, Omule & Brierley, 2005)	
Grasslands	53	Canada	(Rheault, Proulx & Bonin, 2015)	
Meadows—8, 16 & 60 species	340	Germany	(Weigelt et al., 2010)	
Wetlands	182	France, Canada	(Violle et al., 2011, R Proulx et al., unpublished a)	
Notes.

a Data are presented in Appendix 1.

In addition to biomass and height, three of the data sources also included the species richness of their respective plant community (Table S1): meadow, wetland and grassland. This data subset of 533 communities allowed us to examine how expressing standing biomass per unit surface or unit volume modifies the relationship between biomass production and plant species richness.

To determine whether standing dry biomass approximates the volume occupied by plants, we compiled a dataset on 42 wetland plant communities from the Lac St-Pierre ecosystem (Québec, Canada). In August 2014, we sampled vegetation quadrats of 0.5 × 0.5 m covering a broad range of species assemblages and vegetation types. In each quadrat we clipped all the vegetation standing above 2 cm from the ground. We then evaluated plant volume using the water displacement method in a bucket of 12.6 cm radius. We expressed the amount of water displaced by the submersed vegetation in cubic meters. We then oven dried the vegetation at 70 °C for 48 h and weighed the dry biomass. Finally, we report the relationship between plant volume (m3/m2) and standing dry biomass (g/m2) to evaluate their association.

Results

From our analysis of the reviewed data, we found that standing dry biomass per unit surface varied by over three orders of magnitude across the plant kingdom (Fig. 1). In forest ecosystems, dry biomass per unit surface typically ranged between 10 and 100 kg/m2, with pacific and tropical forest communities consistently reaching higher values than that of temperate and boreal forests. In contrast, the standing biomass of croplands and grasslands was comparable to one another, falling just above 1 kg/m2on average. Lastly, meadows cumulated around 0.3 kg/m2 standing dry biomass, although biomass increased with increasing species richness, up to 0.5 kg/m2 in the 60-species communities (slope = 0.0037, R2 = 0.06, df = 338, t = 4.5, p < 0.001).

Figure 1 Standing dry biomass of plant communities in different ecosystems when biomass is expressed (A) per unit surface or (B) per unit volume.

When expressing the standing dry biomass of the 971 communities in our dataset per unit volume, biomass packing varied around 1 kg/m3 (the average 50th percentile being 0.88 kg/m3) and by less than an order of magnitude across ecosystems (Fig. 1). The 75th percentile for biomass packing across ecosystems ranged from 0.81 kg/m3 in croplands up to 1.82 kg/m3 in pacific forests, suggesting high overall consistency relative to measures of biomass per unit surface (Fig. 1). Biomass packing reached an overall maximum at 4.7 kg/m3 in one pacific forest plot.

The positive relationship between plant species richness and biomass production among experimental meadow communities vanished when standing dry biomass was expressed per unit volume rather than surface (slope = 0.0004, R2 < 0.01, df = 338, t = 0.3, p = 0.738, Fig. 1). In other herbaceous ecosystems (Fig. 2), the relationship between plant species richness and biomass production shifted from negative, when biomass was expressed per unit surface (slope = −0.063, R2 = 0.12, df = 191, t = − 5.2, p < 0.001), to nonexistent when expressed per unit volume (slope = 0.018, R2 < 0.01, df = 191, t = 1.3, p = 0.180).

Figure 2 Relationship between species richness and dry standing biomass of plant communities in two herbaceous ecosystems when biomass is expressed (A) per unit surface or (B) per unit volume.

We compared two methods of quantifying the amount of standing vegetation for wetland plant communities. The relationship between dry biomass and plant volume was strong and linear (Pearson’s r = 0.86), indicating that standing biomass is a good proxy for the volume occupied by plant modules, at least for wetland communities (Fig. 3).

Figure 3 Comparison of two methods of assessing the amount of standing vegetation in wetlands of the Lac St-Pierre (Québec, Canada).

The aboveground production of 42 plant communities is expressed in units of dry biomass (g/m2) or of occupied plant volume (m3/m2).

Discussion

Fundamental limits to biomass packing

The literature review highlights that if plant community biomass is represented as dry biomass per unit surface area, major ecosystem types are drastically different. However, when accounting for height and expressing standing dry biomass per unit volume, plant communities from different ecosystems appear remarkably similar in their capacity to pack biomass. Biomass packing across plant communities peaked just below 5 kg/m3, with an average median across ecosystems close to 1 kg/m3. These values are strikingly similar to those originally reported by Weller (1989) for mono-specific stands; reporting a maximum value of 5.2 kg/m3 (median of 1.2 kg/m3) in the first dataset and 5.3 kg/m3 (median of 0.6 kg/m3) in the second. We feel confident that the inclusion of more plant communities would not alter these general results.

Our results revealed that cropland and meadow communities were equally capable of packing biomass per unit volume. However, the caveat to this result is that it is true only when plant communities are compared at their peak of biomass production. For instance, cornfields sown in widely spaced rows attain their maximum of biomass packing only towards the end of the growing season. Likewise, meadows are repeatedly harvested within a year, after which a regrowth period of sparse vegetation and low biomass packing initiates. Moreover, while tropical and temperate forests may show similar packing values at their peak of production, air temperature and sunlight duration impose stronger seasonal variations in the biomass packing of temperate ecosystems. Such examples illustrate the importance of accounting for changes in biomass packing over time and, perhaps, the inaccuracy of treating it as a static metric.

Perhaps the most contentious issue when assessing biomass packing is how the vertical dimension of the sampling box is measured. Here, the upper boundary of that box was defined as the mean height of canopy plants. In our review, one case study did use a physical, rather than a virtual, sampling box to assess the biomass packing of young (<18 years old) forest stands (Peterson, Kabzems & Levson, 1982). In this study, sampling boxes of 1 m3 were placed in 107 woody stands, such that stems would fill the frames from top to bottom. Overall, the maximum biomass packing measured across stands and vegetation types varied from 3 kg/m3 in poplar and birch forests to 6 kg/m3 in pine and spruce forests (Peterson, Kabzems & Levson, 1982). Since sampling frames in this study were systematically placed in the densest portions of the forest, the above values represent upper bounds of dry biomass per unit volume. Hence, although measures of community height among studies might differ slightly, the biomass packing limit of ca. 5 kg/m3 that we identified is in fair agreement with the upper bounds independently reported by Peterson, Kabzems & Levson (1982) and Weller (1989).

Species richness and biomass packing

Garnering much attention in the literature, positive relationships between species richness and biomass production have been repeatedly reported in experimentally manipulated plant communities (e.g., Reich et al., 2012). Further analyses of such experiments have determined that the relationship arises from a more efficient use of resources in species-rich communities over that of species-poor ones, or the so-called complementarity effect (Cardinale et al., 2007; Reich et al., 2012). Plant communities in the Jena Biodiversity Experiment, also used here as a data source, are no exception to this trend (Marquard et al., 2009). In contrast to the positive relationship between species richness and biomass per unit surface, our results show that species-rich meadows in the Jena Experiment do not pack more biomass per unit volume; that is, the richness-packing relationship is flat.

Divergent from those aforementioned biodiversity experiments, a negative relationship between plant species richness and biomass production can be observed in freely assembled herbaceous communities, wherein the productive stands are dominated by fewer species (Waide et al., 1999). The negative tail of the richness-production relationship has been explicated by facilitation and competitive exclusion processes among plant species growing in the most fertile situations (e.g., Michalet et al., 2006). In the present study, we did observe a higher biomass per unit surface in species-poor herbaceous communities, and therefore a negative richness-production relationship, but again the relationship vanished when standing dry biomass was expressed per unit volume. Stated simply, plant species richness had no effect on the biomass packing of plant communities at their peak of production.

To understand why the richness-biomass relationship flattens when expressing biomass per unit volume, we examined how much of the variation in standing biomass per unit surface is explained by community height. Among our 971 plant communities, height explains 88% of the variation in standing biomass when the axes are on a logarithmic scale (Table S1). This strong correlation suggests that both community height and biomass per unit surface are likely driven by similar factors affecting resource use and availability (Moles et al., 2009). Dividing standing dry biomass by community height would emphasize the local species coexistence processes that are independent of stand fertility and climate conditions, but the assertion remains to be fully tested.

Conclusions

Our reanalysis of the literature casts a new light on how plant biomass accumulates in terrestrial ecosystems. Specifically, while the widespread metric of biomass per unit surface highlights differences between plant communities, expressing their standing dry biomass per unit of volume reveals their striking similarity. Expressing biomasses per unit volume goes against a longstanding tradition of measuring primary production per unit surface in terrestrial ecosystems, probably reflecting our utilitarian view of ecosystems conditioned by questions such as: how much crop (e.g., wood, hay, or grain) can be produced and harvested per hectare of land?

Our findings suggest that biomass packing is influenced less by baseline climatic and edaphic conditions, but rather emphasizes local plant–plant interactions. We propose that a fundamental limit might exist to the amount of dry biomass that natural plant communities pack per unit volume in terrestrial ecosystems; since the long-term maintenance of standing dry biomass beyond this point may be biologically unsustainable (Hutchings, 1979). It is likely that this approximate limit of 5 kg/m3 can be exceeded in communities grown under controlled conditions, although croplands did not reach higher biomass packing values over that of other ecosystems in our dataset. Biomass packing may thus represent a simple and generic indicator of resource use efficiency in plant communities.

Supplemental Information

Table S1 Dry standing biomass (g/m2) and height (m) of plant communities from different ecosystems

Click here for additional data file.

Additional Information and Declarations

Competing Interests

Author Contributions

The authors declare there are no competing interests.

Raphaël Proulx conceived and designed the experiments, analyzed the data, wrote the paper, prepared figures and/or tables, reviewed drafts of the paper.

Guillaume Rheault and Laurianne Bonin performed the experiments, contributed reagents/materials/analysis tools, reviewed drafts of the paper.

Irene Torrecilla Roca and Louis Desrochers performed the experiments, reviewed drafts of the paper.

Charles A. Martin analyzed the data, wrote the paper, prepared figures and/or tables, reviewed drafts of the paper.

Ian Seiferling conceived and designed the experiments, performed the experiments, wrote the paper, reviewed drafts of the paper.

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
