# Peer review of "How much biomass do plant communities pack per unit volume?"

_PeerJ, doi:10.7717/peerj.849_

## Round 0.1 · original submission · Major Revisions

· Academic Editor

Major Revisions

Both referees provided excellent feedback. The re-focus provided by
Reviewer 2 in particular is appropriate and will more accurately present the work. If you could explore the ideas proposed by Prof Kikvidze as well that would be fantastic.

·

Basic reporting

No comments

Experimental design

No comments

Validity of the findings

No comments

Comments for the author

Biomass per volume is an interesting measure of plant abundance, although this metrics did not receive due attention so far. This work fills this important gap considerably and thus adds to our knowledge by clarifying two important instants: (1) there are fundamental limits to the amount of standing mass that can be packed per unit volume in terrestrial plant communities, and (2) community composition and species richness do not affect biomass packing.

The text is well written and clear, analytical procedures appropriate, hence the conclusions are valid.

The only question occurred to me is: the aboveground standing mass should be a linear function of volume. Actually, you calculate volume using the height of the standing mass. The product of the height and plant cover also would be an aboveground plant “volume”, and this measure is used, albeit not so often, to quantify plant abundance. I would expect a tight linear relationship between the aboveground dry biomass and the Volume = Cover × Height… Seems to be an interesting hypothesis to test with your data set, if cover data are included in its sufficiently large subset. Then Volume = Cover × Height can be successfully used as a surrogate of the aboveground dry biomass. Your findings might also suggest that this linear dependence is tight and perhaps not dependent on habitat type (e.g., forests versus grasslands). If possible to perform this analyses, could be an interesting addition to your work…

Reviewer 2 ·

Basic reporting

I wrote one general review (below), not in 3 parts.

Experimental design

See below

Validity of the findings

I agreed to review this manuscript based on its title. But I'm afraid it does not live up to this title.

First, describing a pattern does not tell you that it represents some sort of limit. The pattern might be the result of something else. Looking at the population density of humans different places on earth does not answer the question "How many humans can society pack per unit area". Thus, the tile of this manuscript should be "How much biomass do plant communities pack per unit volume?" Similarly, the second sentence in the Abstract should be "Alternatively, expressing production per unit volume allows the comparison of carbon packing by communities".

A more central problem is the definition of volume of a plant community is not as straightforward as this manuscript suggests. The first volume that comes to mind would be the volume according to Archimedes: the volume of water that the community would displace if submerged (or the volume of air it does displace). This would be very difficult to estimate, but extremely interesting. But in this manuscript, volume is defined as the 2 dimensional area covered by the plant community time the height of the plant community. While some plant communities have a clear height, most do not. Is the height of a tropical rain forest the height of most of the canopy, or the height of the emergent trees, which can be MUCH greater? What about boreal forests that do not have a uniform canopy height, but consist of patches of different ages and different heights. The worst case would probably be a savannah. Is the "height" of a savannah the height that of the trees or of the grasses in between?

Even if we restrict ourselves to plant communities that have a relatively clear and homogeneous height and therefore the volume as defined here is meaningful, the ecological implications of this volume are not clear. Why volume should tell us more (or less) than biomass in terms of species diversity is not clear - it is just claimed to be more relevant, without any arguments as to why this should be.

A "thought experiment": Imagine that we could take a plant community, put a giant plate on top of it, and press it down. All plants become shorter and fatter, but are otherwise unaltered. What would this change? It would certainly make things more difficult for MacArthur's warbles, but the implications for diversity and stability of plant communities are not obvious to me. Why does biomass packing "emphasize... species coexistence mechanisms and is an indicator of resource use efficiency in plant communities"? No argument is given.

Throughout the manuscript "production" appears to be used synonymously with biomass, e.g. "...peak of biomass production...". Do the authors mean that maximum biomass has been reached, or that the community is at its fastest growth rate in biomass?

External reviews were received for this submission. These reviews were used by the Editor when they made their decision, and can be downloaded below.

---

## Round 0.2 · accepted · Accept

· Academic Editor

Accept

Thank you for your time. I am accepting this manuscript.